Methods

# Evaluation of the correctable decoding sequencing as a new powerful strategy for DNA sequencing

Chu Cheng, Pengfeng Xiao

**Next-generation sequencing (NGS) promises to revolutionize precision medicine, but the existing sequencing technologies are limited in accuracy. To overcome this limitation, we propose the correctable decoding sequencing strategy, which is a duplex sequencing protocol with conservative theoretical error rates of 0.0009%. This rate is lower than that for Sanger sequencing. Here, we simulate the sequencing reactions by the self-developed software, and find that this approach has great potential in NGS in terms of sequence decoding, reassembly, error correction, and sequencing accuracy. Besides, this approach can be compatible with most SBS-based sequencing platforms, and also has the ability to compensate for some of the shortcomings of NGS platforms, thereby broadening its application for researchers. Hopefully, it can provide a powerful new protocol that can be used as an alternative to the existing NGS platforms, enabling accurate identification of rare mutations in a variety of applications in biology and medicine.**

## Introduction

Since the creation of the "dideoxy chain termination reaction sequencing method" by Sanger at the University of Cambridge in 1977, DNA sequencing has become one of the routine methods of modern biological research. Sanger sequencing, which belongs to the first-generation sequencing, has made important contributions to our understanding of genome diversity in health and disease. However, because of the limited throughput and high cost of this technology, the next-generation sequencing (NGS) platforms were developed. NGS technologies have greatly reduced the cost of human genome sequencing (from $100,000,000 to $1,000), and have had a huge impact on the research in contemporary biology, medicine and other fields (1, 2). NGS platforms have become the current mainstream sequencing platforms, enabling the use of sequencing as a clinical tool and providing one of the main sources of medical big data (3, 4, 5).

NGS platforms provide a large amount of data, but the error rate (~0.1–15%) is higher (6) than that of the traditional Sanger sequencing platform (error rate of 0.001%) (7, 8). Although high-coverage assembly can reduce sequencing errors, it only guarantees the accuracy of sequencing information for a certain abundance sequence, and low-abundance sequences may be discarded as sequencing errors. Therefore, when the same DNA template can be sequenced multiple times in different ways (not simple repetitions), and the sequencing information can be completely aligned, the accuracy of information from a single read can be evaluated. In the previous study, both Pu (9) and Chen (10) conducted sequencing-by-synthesis (SBS) method by adding a set of dual-base (AG/CT, AC/GT, or AT/CG) to each reaction. However, the existing dual-base addition sequencing technique fails to solve the problem of homopolymer sequencing and it may also introduce a longer homopolymer (e.g., in AC/GT dual-base addition, information for sequence fragments such as TTTGGGGGTGTTGGT, AAACCAACCCA, etc.), thereby potentially producing more errors than traditional single-nucleotide addition. As a result, the high error rate of the original data makes it difficult to judge the information from a single read.

To address this problem, we proposed a correctable decoding sequencing technology based on dual-nucleotide SBS. This strategy applies a mixture of two types of nucleotides, natural nucleotide (denoted as X) and cyclic reversible termination (CRT) (denoted as Y*), to interrogate a template in two parallel sequencing runs. The 3'-OH groups of CRTs have been blocked, and hence, after the nucleotide is incorporated onto the complementary synthetic strand, the strand will not be further extended (11). By using this synthetic characteristic of CRTs, when N nucleotide synthesis occurs in this sequencing reaction, the information for this sequencing reaction is (N-1) specific base X and an encoding (XY) with partially defined base composition. Thus, a large number of specific bases and encodings can be obtained by only a single sequencing run. When the template is sequenced twice with different types of added dual-nucleotide, two sets of such sequencing and encoding information are obtained sequentially, thereby base sequence can be accurately deduced. This strategy can eliminate or significantly reduce the sequencing error of homopolymer, and greatly improve sequencing accuracy.

---

State Key Laboratory of Bioelectronics, School of Biological Science and Medical Engineering, Southeast University, Nanjing, China

Correspondence: xiaopf@seu.edu.cn

Here, we discuss the potential advantages of this technology in terms of sequencing accuracy, sequence decoding and reassembly. Through simulation we are able to build an effective strategy to correct the sequencing errors, and eventually improve raw accuracy. Besides, we also discuss the current challenges of this correctable decoding sequencing for NGS, and its possible applications to current sequencing platforms. We hope it will provide a new sequencing protocol to break through the bottleneck of current NGS platforms in confirming low-abundance sequences, which has important application value in clinical diagnosis of early disease markers.

# Results

### Mechanism of the correctable decoding sequencing approach

In general, natural nucleotides (A, T, C, and G) and CRTs (A*, T*, C*, and G*) can form six sets of dual-nucleotide additions (AT*/CG*, AC*/GT*, AG*/TC*, AC*/TG*, AG*/CT*, and AT*/GC*), corresponding to (GA*/TC*, TA*/CG*, CA*/GT*, GA*/CT*, TA*/GC*, and CA*/TG*). This technology applies any two of the six sets of dual-nucleotide additions to interrogate the template in two parallel runs. Because the signal intensities of released identical detection molecules (such as pyrophosphates ([12]), H$^+$ ([13]), fluorescent molecules ([14], [15]) etc.) are proportional to the number of incorporated natural nucleotides or/and CRTs, two sets of encodings, which contain information about the possible types and numbers of incorporated base(s) in each cycle, can be acquired. For example, when dual-nucleotide addition AT*/CG* is used (Fig 1A), in the first extension reaction (AT*), one dA is paired with the first base (T) and generate one signal intensity, then the reaction stops upon the second base G because of the base mismatch. In the following extension reaction (CG*), one dC and one dG* are paired with the next two bases (GC) and yield two signal intensities, then stops upon the fourth base G because of the blocked 3′-OH group of G*. The 3′-OH is regenerated with tris(2-carboxyethyl) phosphine (TCEP) after two extension reactions.

The amount of signal intensity produced in each extension reaction is equal to the number of incorporated nucleotides. We use a two-digit code "$NM$" to represent the number of nucleotides added in a single sequencing cycle. Conjugated mixture AT* and CG* are introduced alternately to react with the DNA template primed with the starting sequence TGCGAA (Fig 1B). $N^1 = 1$, $M^1 = 2$, means that only one nucleotide synthesis in the first extension reaction and two nucleotides are incorporated in the second extension reaction. It can be inferred that the first base must be A, because the 3′-end of the synthesized strand is not terminated by T*, which can be continuously extended ($M^1 > 0$). In addition, $M^1 = 2$ can be transformed to an explicit base C and an encoding (CG), which means C or G. After the 3′-OH is regenerated with TCEP, another sequencing cycle is started. For the second sequencing cycle, $N^2 = 0$ means that no nucleotide synthesis reaction occurs, and $M^2 = 1$ can be converted as an encoding (CG). Moreover, from $N^1 = 1$, $M^1 = 2$, $N^2 = 0$, $M^2 = 1$, it can be concluded that the former encoding (CG) must be G because AT* and CG* have already provided an opportunity for the synthesis of A, G, C, and T, and this situation will only occur if the synthesis chain is terminated by G*.

In this way, a set of two-digit strings ($N^1M^1$, $N^2M^2$, $N^3M^3$, ..., $N^kM^k$) is obtained sequentially through $K$ cycles in a sequencing run. It is assumed that conjugated mixes XY* and WZ* are alternately introduced to react with the template in each sequencing cycle, and a two-digit string $N^iM^i$ is obtained in $i$ cycle. The decoding algorithm that converts the two-digit strings into base-encoding is as follows:

(1) if $N^i > 0$, $M^i = 0$, $i = 1,2,...,k−1$, there are $N^i − 1$ base(s) X and one base Y.
(2) if $N^i > 0$, $M^i = 0$, $i = k$, there are $N^i − 1$ base(s) X and an encoding (XY).
(3) if $N^i ≥ 0$, $M^i > 0$, $i = 1,2,...,k$, there are $N^i$ base(s) X, $M^i − 1$ base(s) W and an encoding (WZ).
(4) if $N^i ≥ 0$, $M^i > 0$, $N^{i+1} = 0$, $M^{i+1} > 0$, $i = 1,2,...,k−1$, there are $N^i$ base(s) X, $M^i − 1$ base(s) W and one base Z.

Therefore, the set of two-digit strings "12 01 10 10 01 01 01 01 20 02 01 32" obtained by the first sequencing run can be translated to the sequencing information ACG(CG)TTGGG(CG)ATCG(CG)AAAC(CG). For the second sequencing run, another two-digit string "10 01 11 02 50 02 11 10 10 10 01 10" is generated, and can be translated in the same way into the sequencing information A(CT)GCT(CT)GGGGATCG-CAAA(CT)(AG) (Fig 1C). By aligning these two sets of encodings sequentially, the sequence can be accurately deduced. Therefore, the complementary sequence of the template is 5′-ACGCTTGGG-GATCGCAAACG-3′.

Using our own designed software that encodes the sequencing information by a four-color code, the explicit base information can be deduced (Fig 1D). A two-color code means an ambiguous base, whereas a one-color code represents an explicit base. In addition, the number of single-color and two-color code represents the number of incorporated bases. When the template is sequenced by dual-nucleotide addition, the corresponding color codes are shown in Fig 1D. The sequence can be deduced by comparing the same color code between the two compared two-color codes.

### The correctable decoding sequencing approach reduces the complexity of sequence decoding and reassembly

In general, the existing dual-base sequencing technology cannot obtain explicit bases in a single sequencing run, increasing the workload of sequence decoding and assembly. As we known, obtaining accurate single read information can reduce the complexity of sequence decoding and reassembly, thereby decreasing the coverage required for a complete sequence, and reducing the cost of sequencing.

We randomly generate 20 different DNA template sequences with length 50 bp (Table S1) and simulate the correctable decoding sequencing reactions. The results reveal that no matter which dual-nucleotide addition is used, using this technology, 70–80% of the calls of a single sequencing run are unambiguous (Fig 2A). By calculating the average, it can be concluded that 74% of the explicit bases can be obtained in a single sequencing run, making decoding substantially less effort (Fig 2B). In addition, unlike NGS, the template is interrogated by two different sequencing runs with this approach, so that single read information accuracy can be ensured by aligning two sets of four-color codes. Thus, this technology can

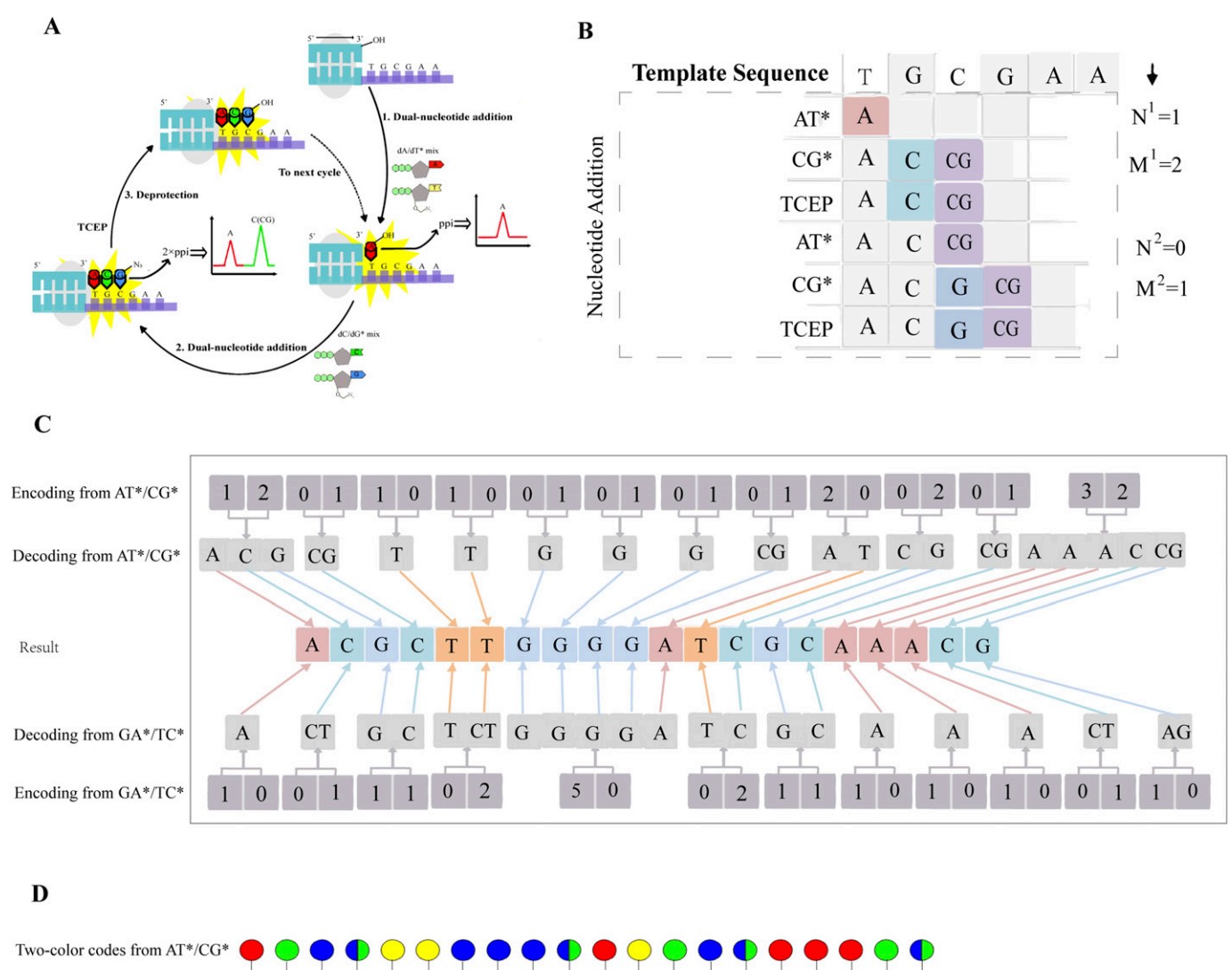

**Figure 1. Mechanism of the correctable decoding sequencing approach.**
**(A)** Each sequencing cycle consists of nucleotide extension, signal detection and deprotection. **(B)** A sequence is interrogated using AT*/CG* in a single sequencing run, and a set of two-digit string, which can be converted into base information, is obtained. **(C)** The decoding scheme of this approach. The two bases in the same box represent an ambiguous base. **(D)** Four-color codes from dual-nucleotide addition and the procedure for decoding by the simulation software.

assemble the genomic sequences by smaller multiplier coverage. In conclusion, the correctable decoding sequencing approach can reduce the complexity of sequence decoding and reassembly, and it can undoubtedly decrease the cost of high-throughput DNA sequencing.

Moreover, because the correctable decoding sequencing approach has the function of judging whether the single read sequencing information is correct, it provides the possibility of valid confirmation of low-abundance sequences information. Therefore,

we believe that this approach has great scientific significance for finding early markers of phenotype at the molecular level, and also has important application value for early clinical diagnosis.

### Error correction strategy of the correctable decoding sequencing approach

In high-throughput DNA sequencing, accuracy is a very important indicator to measure sequencing strategies. For the current dual-

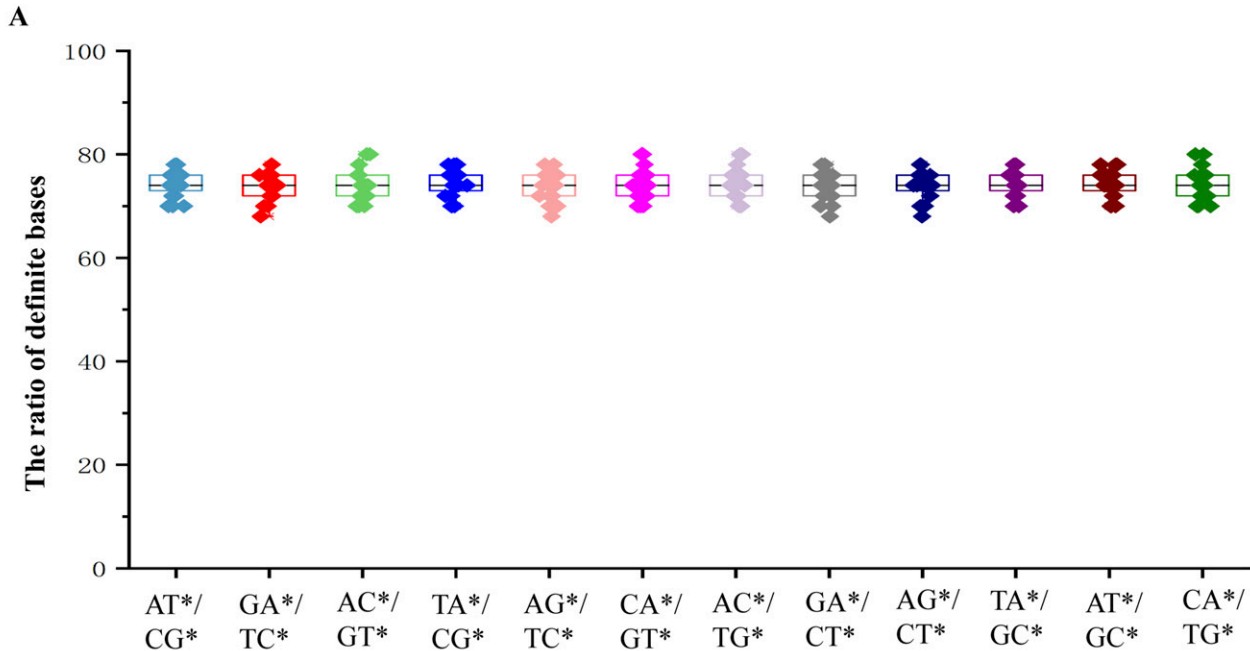

Figure 2. **The ratio of definite bases in a single sequencing run obtained by the correctable decoding sequencing.**
**(A)** The definite base ratio distribution of different templates in a single sequencing run. **(B)** The average value of the definite base ratio obtained by different dual-nucleotide additions.

base addition sequencing technology, the theoretical foundation relies on a proportional increase in the signal as multiple nucleotides are incorporated. However, homopolymer regions are difficult for these sequencing platforms, which lack sequencing accuracy in measuring homopolymers larger than 6 bp ([16], [17]), ultimately leading to a high sequencing error rate.

The correctable decoding sequencing approach can effectively solve the problem of homopolymer sequencing. When signal intensities obtained by a single sequencing run are not linearly proportional to the number of polymer bases, this situation can be defined as an ambiguous number of homopolymer, and this region must be clear because of base-by-base nucleotide incorporation in the other sequencing run. Therefore, ambiguous alignment can be used to align these two sets of four-color codes by previously designating a dynamic range of base number. For example, when template T1 (Table 1) is sequenced by the dual-nucleotide addition AT*/CG* and GA*/TC*, respectively, two sets of two-color codes, S1 and S2, are obtained (Fig 3A). For the homopolymer region, S1 has clear encodings from base-by-base measurement, but S2 has an ambiguous number of this fragment. Through dynamic programming, the homopolymer segment is

filled in and the remaining parts are used to deduced the base sequence accurately.

The correctable decoding sequencing approach can also efficiently rectify sequencing errors. As is the case in Fig 3B, the original $N^4 = 6$ is mistakenly measured as three in cycle 4 of the second sequencing run and causes an error, resulting in failure to align the sequencing information from two sets of four-color codes. In fact, the sequencing errors, which occurs in one run, do not affect subsequent sequencing results. The DNA sequence can be decoded by right-shifting three bits since the 10th bit, and then an error-free sequence is obtained. However, for the existing dual-base sequencing technology, the two sets of four-color codes can be aligned even if there are sequencing errors in the original data, and therefore, the wrong base information is decoded. This means that correctable decoding sequencing approach can effectively detect errors and rectify them by changing the corresponding two-digit string based on the context. Errors must be rectified sequentially from the first error because a change in a two-digit string corresponding to the shift operation, will affect downstream decoding.

To demonstrate the robustness of this approach, 100 DNA template sequences with length 100 bp are generated (Table S2).

**Table 1.  Sequences used in the assay[a].**

| Template | Sequence (5′-3′) |
| --- | --- |
| T1 | CTTGTATAGTGACGAGCGTTAGAAGGCCGTATAATCGCAACCTTTACGCCCCCCTAGACC CTACGATGGAACTAAGTCTA |
| T2 | CTTGTATAGTGACGAGCGTTAGAAGGCCGT[A/G]TAATCGCAACCTTTACGCCCCCCTAGACC CTACGATGGAACTAAGTCTA |
| T3 | CTTGTATAGTGACGAGCGTTAGAAGGCCGTATAATCGCAAC[G]CTTTACGCCCCCCTAGACC CTACGATGGAACTAAGTCTA |
| T4 | CTTGTATAGTGACGAGCGTTAGAAGGCCG[-]ATAATCGCAACCTTTACGCCCCCCTAGACC CTACGATGGAACTAAGTCTA |
| SP | TAGACTTAGTTCCATCGTAG |

[a]T1–T4 represent the templates used. SP represents sequencing primer. The underlined segments are the hybridization regions with the sequencing primer SP. The segments in the bracket are SNP/deletion/insertion.

In the simulation, we randomly introduce 1%, 2%, 3%, and 5% sequencing errors, respectively, into the sequences to calculate the sequencing accuracy. The results show that the correctable decoding sequencing approach can eliminate the most raw sequencing errors. When the raw error rate is below 2%, almost all errors can be completely rectified after correction (Fig S1). Therefore, this approach is capable of accurately identifying these errors and assembling the correct sequence information.

In addition, it should be noted that when the template is interrogated with the same sequencing cycle, the number of four-color codes obtained from two parallel runs may not be exactly the same. As Fig 3C shows, the number of four-color codes obtained from two parallel runs is different when the template is interrogated with 36 sequencing cycles. Error correction can be performed according to the short four-color codes, and then the remaining four-color codes are filled in. In this way, on the one hand, the obtained information can be directly used as a read for sequence assembly; on the other hand, further error correction can be performed by increasing the coverage of the assembled sequence, to further improve sequencing accuracy.

### The correctable decoding sequencing approach improves sequencing accuracy

As for sequencing accuracy, data quality has nothing to do with the number of times that template sequencing is performed (18, 19), and when the error rate of multiple sequencing is constant, the reduction in the error rate is determined by the square of the error rate of a single sequencing run (20). In the correctable decoding sequencing strategy, the template needs to be interrogated by two parallel sequencing runs. Therefore, after correction, only when the same sequencing error occurs twice at the same position can a completely aligned sequence be obtained. According to the existing NGS platforms, 454 platform can provide seven types of specific signal information (0, 1, 2, 3, 4, 5, ≥6) for each sequencing reaction (16, 17), whereas other platforms can accurately determine the information for eight bases (13), thus providing nine types of information (0, 1, 2, 3, 4, 5, 6, 7, and ≥8). In this study, we use $N$ to denote the amount of information that can be provided in each sequencing reaction, and $R$ to denote the error rate of a single sequencing run. In theory,

$$P = R^2 \times [1/(N-1)]^2,$$

where $P$ is the theoretical value of the error rate of the correctable decoding sequencing strategy.

Assuming $N$ = 7, the functional relation between $R$ and the logarithm of $P$ can be obtained (Fig 4). From Fig 4, we find that $P$ decreases exponentially with $R$. Therefore, assuming a conservative value of $R$ = 1.8% (454 Roche 1% (21), Illumina 0.26-0.8% (2), Ion Torrent 1.78% (22)), and $N$ = 7, thus $P$ can be approximated as: 1.8% × 1.8% × 1/6 × 1/6 = 0.0009%. This calculated error frequency is lower than that for Sanger sequencing, making the proposed approach has the potential to be the most accurate sequencing technology.

### Detection of SNP/insertion/deletion by the correctable decoding sequencing

When no error is believed to be present in the sequencing reaction, it is impossible to determine whether SNP, insertion or deletion occurs in the sequence. This requires the introduction of reference sequences for follow-up data analysis. Generally, a reference sequence can be obtained in two ways: one is the genome sequence that has been sequenced; the other is the sequence used as the reference sequence for high-coverage sequencing. The detailed process includes: (i) translation of the reference sequence into two sets of four-color codes by software; (ii) comparison of the four-color codes of the reference sequence with those from the sequencing information for alignment with mapping algorithm.

We use template T2-T4 to simulate the process of determining SNP/insertion/deletion with the correctable decoding sequencing approach (Fig 5). The four-color codes of template T2 and its alignment with corresponding references are shown in Fig 5A. Comparing S1 with the reference sequence R1, only one mismatched site appears in the 30[th] bit and the other sites can be matched perfectly. Moreover, the other set of four-color codes exactly match the reference sequence (S2 versus R2). There must be a SNP in the sequence. When a mismatched site is found, and most of the subsequent four-color codes cannot fully match compared S1 and S2 with references R1 and R2, this indicates that an insertion or deletion may have occurred. In Fig 5B, a mismatched site is identified in S1 and S2 at the 20[th] bit and S1 and S2 exactly match the reference sequence R1 and R2, respectively, by left-shifting one bit since the 20[th] bit, base G must be inserted in the queried sequence. In Fig 5C, comparing S1 and S2 to references R1 and R2, the two sets of four-color codes can be perfectly aligned by right-shifting one bit since the 31[st] bit, there must be base T deleted from the sequence. Therefore, alignment tools can be developed to automatically map both sets of four-color codes to detect SNP/insertion/deletion quickly and efficiently.

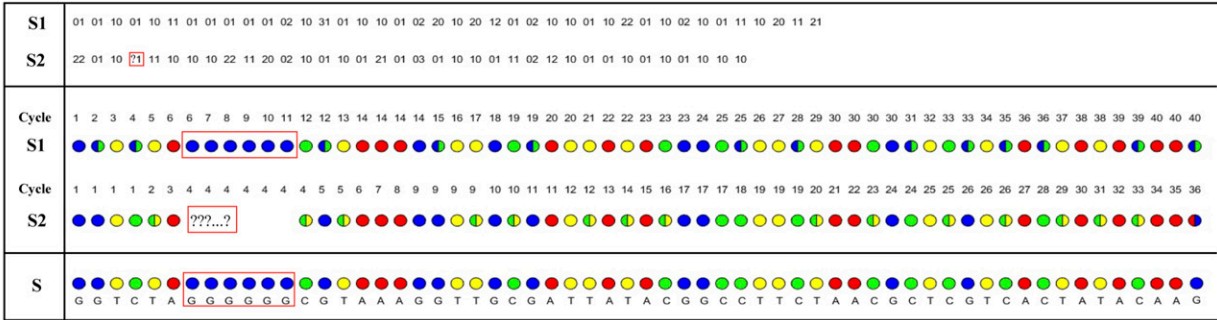

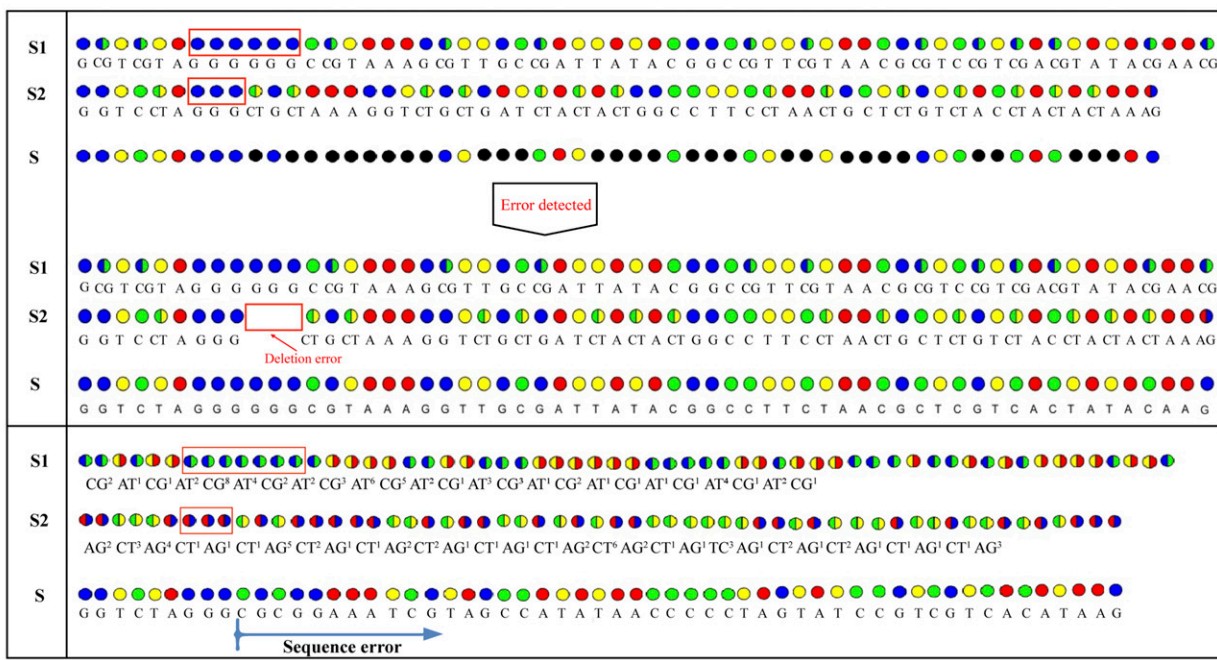

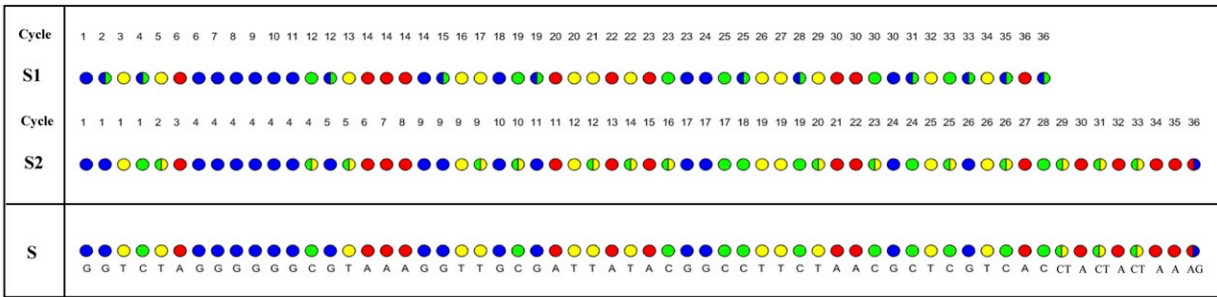

**Figure 3. Strategy for error correction.**
The two-digit strings and four-color codes from interrogating template T1 by the correctable decoding sequencing technology. S1 and S2 represent the four-color codes obtained from AT*/CG* and GA*/TC*, respectively. **(A)** Strategy for homopolymer regions. **(B)** Comparison of the correctable decoding sequencing technology with the existing dual-base addition sequencing technology for error correction. The black dot means mis-decoded. **(C)** Error correction and sequence assembly for different number of four-color codes.

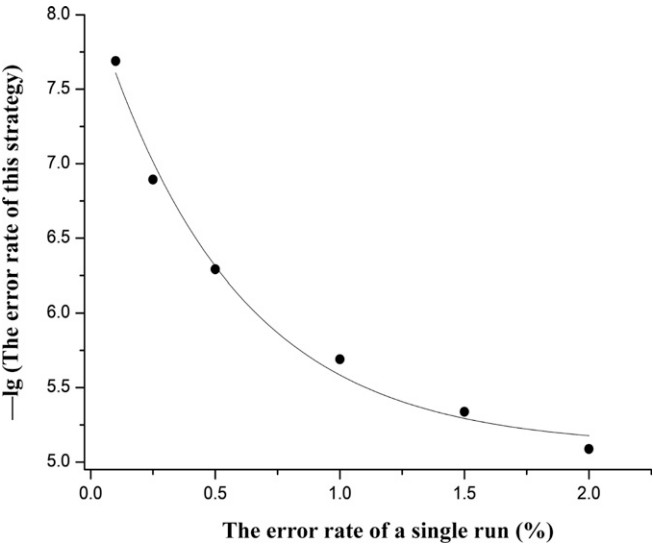

**Figure 4.** Functional relation between the error rate of a single sequencing run and the logarithm of the error rate of this strategy.

For the correctable decoding sequencing technology, the same DNA template can have two sets of four-color codes that can be used for alignment. Therefore, the information available for comparison provides twice the coverage of general sequencing methods. As a result, this technology has higher efficiency and accuracy when detecting SNP/insertion/deletion. Moreover, because this technology has the function of determining whether the information from a single read is correct, SNP/insertion/deletion can also be detected by the comparing individual samples. Therefore, this approach has great advantages for determining low-abundance sequences and provides an effective analysis tool for detection of early gene mutations.

### Differences in similar sequence can be amplified by the correctable decoding sequencing approach

When differentiating species, a certain region is usually selected as the target. Some regions differ greatly between species in both nucleotide composition and size, whereas others are conserved for retained enzymatic activity. When the similarity of the strains sequence is very high (e.g., a single nucleotide difference), it is difficult to distinguish them by conventional sequencing technology. According to the principle of the correctable decoding sequencing technology, the start position of sequencing will not affect the encoding of a given region. Therefore, this approach can be used for species differentiation.

Here, we choose *Streptococcus* species and the variable P3 region of *rnpB* gene to stimulate the process of species identification. The target regions, consisting of the P3 regions and four nucleotides downstream of the P3 region in four *Streptococcus* species (*Salmonella infantis*, *Streptococcus peroris*, *Streptococcus anginosus*, and *Streptococcus constellatus*), are shown in Table 2.

The sequencing results for the P3 region of four *Streptococcus* species are predicted from a set of dual-nucleotide addition,

GA*/TC*, using self-developed decoding software (Fig 6A). The signal intensity distribution is shown in a histogram (Fig 6B). As can be seen from Fig 5A and B, the number of incorporated nucleotides is identical from the 1st to the 18th cycle. However, in the 19th cycle, one and two bases are incorporated for the two species, respectively. *S. infantis* and *S. peroris* incorporate only one nucleotide in the 19th cycle, whereas *S. anginosus* and *S. constellatus* incorporate two nucleotides. In addition, as for *S. anginosus* and *S. constellatus*, the number of incorporated nucleotides in the 21st cycle is different. Compared with the number of added bases in the 28th cycle, *S. infantis* incorporates two bases, whereas *S. peroris* incorporates only one base. Thus, these four species can be distinguished from each other by only a single sequencing run.

The similarity of the strain sequences in the P3 region is high, and the sequence difference displayed is low (deletion or insertion or substitution of one nucleotide). Such few differences need to be considered for further evaluation to determine the specific classification. Therefore, it is arduous to distinguish a single nucleotide difference in the P3 region of *S. infantis* or *S. peroris* by traditional pyrosequencing technology (23). However, the correctable decoding sequencing approach can successfully distinguish between these two species, *S. infantis* and *S. peroris*. The number of incorporated nucleotides in each cycle is found to be different since the 28th cycle. The sequence differences among species can be amplified by this strategy, making separation and identification more accurate and reliable. Thus, this approach has great error-tolerant potential in distinguishing biological variation from sequencing errors. In addition, there is no need to reveal specific sequence information, and the templates need to be sequenced only once.

### Possible applications to current sequencing platforms

According to the principle of the correctable decoding sequencing technology, it can be expected to be compatible with most SBS-based sequencing platforms, such as pyrosequencing device, 454 system, Ion Torrent, and ECC sequencing platform.

Pyrosequencing device is based on the theory that when a dNTP is incorporated into a DNA strand, a bioluminescence signal generated by a cascade of enzymatic reactions can be detected by a charge-coupled device camera (24). Because the current pyrosequencing technology uses natural nucleotides to analyze PCR products, which limits its application. Zhou et al. introduced ddNTPs into pyrosequencing (25). This enabled the analyzed template not to generate sequencing signal, which made possible the development of new analysis methods and new application areas for the analysis of multi-template PCR products. For example, in the analysis of multiple SNP sites because each site has the possibility of two homozygous types and one heterozygous type, $3^N$ separate profiles are required when analyzing a DNA mix template containing N SNPs at a time by the existing analytical methods for specific maps (26), which makes it difficult to analyze more than three SNP sites. In the correctable decoding sequencing technology, CRT can be used to replace ddNTPs in pyrosequencing, so that simpler operations and lower analysis costs, as well as wider application can be achieved for complex analysis.

## A

### SNP

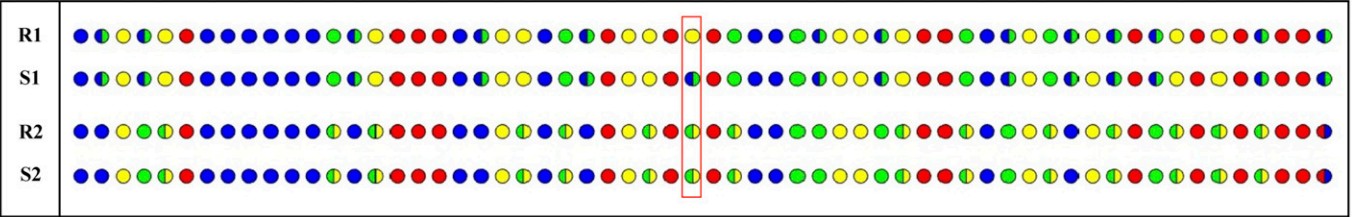

## B

### Insertion

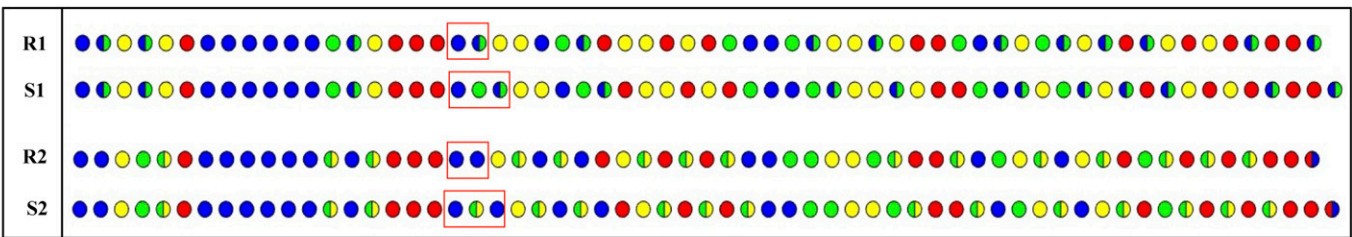

## C

### Deletion

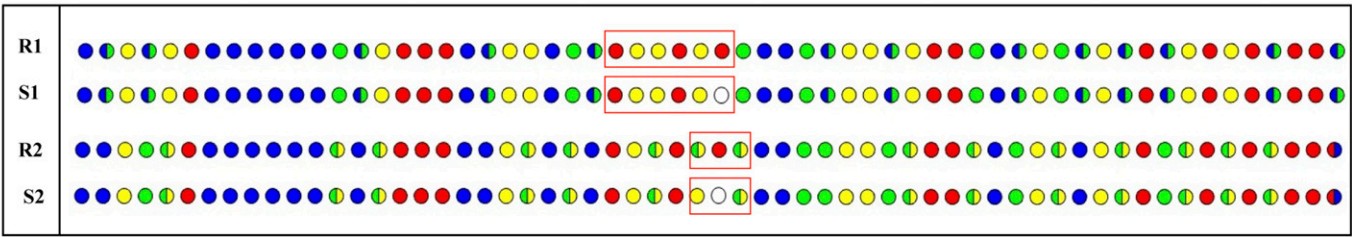

**Figure 5. Detection of SNP/insertion/deletion by the correctable decoding sequencing.**
S1 and S2 represent the four-color codes obtained by interrogating templates T2-T4 with AT*/CG* and GA*/TC*, respectively. R1 and R2 indicate the four-color codes obtained from the reference by using AT*/CG* and GA*/TC*, respectively. **(A)** A SNP, A/G is contained in template T2. **(B)** Insertion base G is contained in template T3. **(C)** Deletion is contained in template T4. The four-color codes in the red boxes indicate differences between the original data and the reference.

454 system is the first NGS instrument, which uses emulsion PCR and pyrosequencing technology (27, 28). Therefore, the correctable decoding sequencing technology is compatible with 454 system. Although the 454 system offers superior read length, it has a major limitation with regard to homopolymer regions because of its lack of single-base accuracy in measuring homopolymers larger than 6–8 bp (16, 17). Therefore, when the correctable decoding sequencing technology is used on this platform, the problem of homopolymer sequencing can be fully addressed, fundamentally improving data quality.

Ion Torrent is the NGS platform that uses semiconductor. Rather than detecting light signal, the Ion Torrent platform monitors the pH to recognize whether the dNTP is incorporated or not (5, 29). Much as in 454 system, the pH change detected by the sensor has poor linearity with respect to the number of nucleotides incorporated in a single reaction cycle, limiting accuracy in measuring

**Table 2. Target region consisting of the P3 regions and four nucleotides downstream of the P3 region in each *Streptococcus* strain.**

| Isolates | Sequence (5′-3′) |
|---|---|
| *S. infantis* | CGTGGAGAGTTTATCTTTTCATGA |
| *S. peroris* | CGTGGAGGGTTTATCTTTTCATGA |
| *S. anginosu* | CGTGAAGAGTTCGTCTTTTCATGA |
| *S. constellatus* | CGTGAAGAGTCGTCTTTTCATGA |

homopolymer region. Therefore, the combination of the correctable decoding sequencing and Ion Torrent can be expected to further improve sequencing accuracy, making the combination much more useful for applications.

ECC sequencing is the NGS platform based on a dual-base combined with fluorogenic SBS proposed by Chen et al. (10). This

**A**

| | GA* | TC* | GA* | TC* | GA* | TC* | GA* | TC* | GA* | TC* | GA* | TC* | GA* | TC* | GA* | TC* | GA* | TC* | GA* | TC* | GA* | TC* | GA* | TC* | GA* | TC* | GA* | TC* | GA* | TC* | GA* | TC* | GA* | TC* | GA* | TC* | GA* | TC* |
|---|---|---|---|---|---|---|---|---|---|---|---|---|---|---|---|---|---|---|---|---|---|---|---|---|---|---|---|---|---|---|---|---|---|---|---|---|---|---|
| *S.infantis* | 0 | 2 | 1 | 0 | 0 | 1 | 2 | 0 | 1 | 0 | 1 | 0 | 1 | 0 | 2 | 0 | 0 | 1 | 1 | 0 | 1 | 0 | 1 | 0 | 0 | 1 | 0 | 2 | 0 | 2 | 0 | 1 | 1 | 0 | 0 | 1 | 1 | 0 |
| *S.peroris* | 0 | 2 | 1 | 0 | 0 | 1 | 2 | 0 | 1 | 0 | 1 | 0 | 1 | 0 | 2 | 0 | 0 | 1 | 1 | 0 | 1 | 0 | 1 | 0 | 0 | 1 | 0 | 1 | 0 | 1 | 0 | 2 | 0 | 1 | 1 | 0 | 0 | 1 | 1 | 0 |
| *S.anginosus* | 0 | 2 | 1 | 0 | 0 | 1 | 2 | 0 | 1 | 0 | 1 | 0 | 1 | 0 | 2 | 0 | 0 | 1 | 2 | 0 | 1 | 0 | 0 | 1 | 0 | 2 | 0 | 3 | 1 | 0 | 0 | 1 | 1 | 0 | | | | |
| *S.constellatus* | 0 | 2 | 1 | 0 | 0 | 1 | 2 | 0 | 1 | 0 | 1 | 0 | 1 | 0 | 2 | 0 | 0 | 1 | 2 | 0 | 0 | 1 | 0 | 2 | 0 | 3 | 1 | 0 | 0 | 1 | 1 | 0 | | | | | | |

**B**

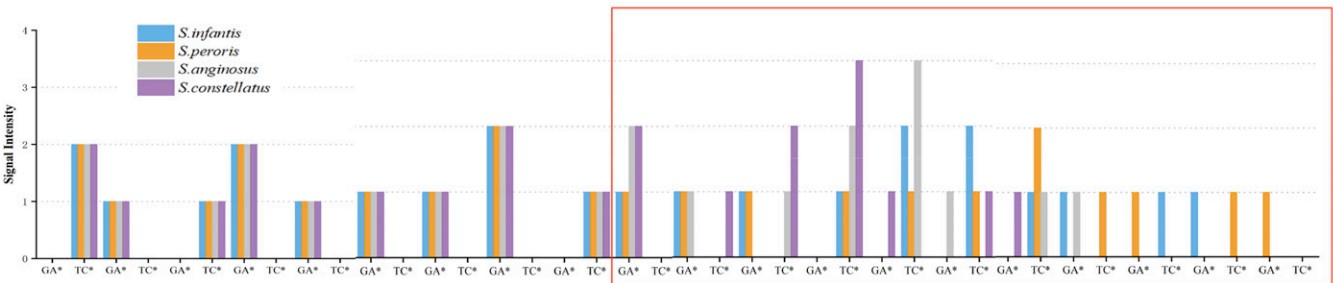

**Figure 6.  Dual-nucleotide addition with GA*/TC* for the differentiation of four *Streptococcus* species.**
**(A)** The sequencing results for the P3 region of four *Streptococcus* species from a set of dual-nucleotide addition, GA*/TC*. **(B)** Histogram of the signal intensity distribution of the four species. The red box indicates differences among these strains.

approach allows the introduction of a mixture of two types of nucleotides with free 3′-OH group into each reaction cycle. When the DNA template is interrogated three times, three sets of encoding can be obtained, from which an unambiguous sequence can be accurately deduced. However, this dual-base sequencing approach fails to solve the problem of homopolymer sequencing and may also introduce a longer homopolymer, thereby potentially leading to more errors than traditional single-nucleotide addition. According to their report that the accuracy of the original data is only 98.1%, which is the lowest among the NGS platforms (454 system 99% (22), Illumina 99.2–99.78% (2), Ion Torrent 98.22% (24)). Therefore, if ECC sequencing is combined with the correctable decoding sequencing, only two sequencing runs need to be carried out, and sequencing accuracy may well be improved.

### Current challenges of the correctable decoding sequencing approach for NGS

In this correctable decoding sequencing approach, a mixture of natural nucleotide and CRT is added in each extension cycle. CRT is a kind of nucleotide analogue in which the ribose 3′-OH group is modified with a reversible chemical moiety to temporarily terminate the polymerase reaction. At present, commonly used CRTs in sequencing include 3′-O-$N_3$-dNTPs (30), 3′-O-allyl-dNTPs (31), 3′-O-(2-nitrobenzyl)-dNTPs (11) etc. Although CRT provides advantages in homopolymer sequencing, the imperfection of reaction chemistries, such as scars left on the nascent strand after cleaving the reversible chemical moiety, limit the read length (32). In general, as for SBS, the read length is related to the number of reaction steps in a single sequencing cycle and the type of incorporated nucleotides, and an increase in the number of steps will make the sequencing

cycle inefficient (33). The functional relationship between read length and cycle efficiency (Ceff) is as follow: $(Ceff)^{read\ length} = 0.5$ (3). Therefore, the additional step (such as deprotection of CRTs) may affect Ceff, which in turn affects the read length. In principle, solely in terms of read length, choosing natural nucleotides as raw materials for synthetic sequencing reactions has more advantages than using modified nucleotides. Therefore, using a mixture of natural nucleotide and CRT for the synthesis reaction cannot only take advantage of the homopolymer sequencing accuracy provided by CRT, but also reduce its impact on read length to a certain extent.

The correctable decoding sequencing approach can be used to detect transient bioluminescent or electrochemical signals that require continuous monitoring. This means that each individual sequencing reaction requires a separate microreactor and detector, which affects the throughput. In addition, a single sequencing cycle in this approach consists of nucleotide extension, signal detection and deprotection, and therefore some signal intensity decay is inevitable. The decay, which is mainly due to the loss of bead-base templates during washing, has caused challenges in base-calling. One way to improve this situation is to implement automatic control of the base addition and deprotection cycle using microfluidics. Therefore, when this technology is used on the current platforms, some modifications need to be made to accommodate it, which may lead to an increase in sequencing costs.

## Discussion

The birth of NGS technology is a milestone event in the field of biomedical research, and its application range is very wide. Through

this technology, researchers can obtain the most comprehensive view of genomic information and related biological implications (34). However, as far as error rate is concerned, Sanger sequencing (error rate of 0.001%) is still the gold standard (21, 35). Therefore, in clinical application, the results obtained by NGS platforms still need to be confirmed by Sanger sequencing. The correctable decoding sequencing technology proposed in this article consigns the major drawback of high error rate in NGS to history, with a conservative theoretical error rate of 0.0009%, which is lower than Sanger sequencing.

As is well known, repetitive DNA sequences are abundant in bacteria and mammal, and human genomes, and homopolymer inaccuracy prevents wider use of NGS (36). Based on the principle of the correctable decoding sequencing technology, every homopolymer can be extended exclusively in at least one of the two sequencing runs. Thus, this technology is much less susceptible to homopolymer errors when determining the length of homopolymers. Considering the supremacy of this technology in terms of sequencing accuracy, we are optimistic that it would contribute to various applications, including rare mutation detection and early biomarker identification.

Unlike the existing NGS technologies, the template is interrogated via multiple parallel sequencing runs (not simple repetitions) with the correctable decoding sequencing technology, we can judge whether the single read sequencing information is correct. Therefore, this technology can overcome the limitation of NGS for low-abundance mutations, and provide the possibility of valid confirmation of low-abundance sequence information, which is important for precision medicine.

Moreover, unlike the previously proposed dual-base method, about 74% of explicit bases can be obtained in a single sequencing run with the correctable decoding sequencing, which makes decoding substantially less effort. Thus, this technology can fundamentally improve data quality, and the accurate single read information can reduce the complexity of sequence decoding and reassembly, thereby decreasing the coverage required for a complete sequence, and undoubtedly reducing the cost of sequencing.

Another attractive advantage of the correctable decoding sequencing technology is its compatibility. In theory, it is compatible with the sequencing platforms based on the linear relationship between the released molecules and the number of incorporated nucleotides, such as pyrosequencing, 454 system, Ion Torrent, and ECC sequencing platform etc. Moreover, it also has the ability to compensate for some of the shortcomings of NGS platform, thereby broadening its application for researchers. Therefore, the correctable decoding sequencing technology has the potential to provide a powerful new protocol that can be used as an alternative to current and upcoming sequencing platforms, enabling accurate identification of rare mutations in a variety of applications in biology and medicine.

## Data Availability

The data that support the findings of this study are available from the corresponding authors on reasonable request.

## Supplementary Information

## Acknowledgements

This study was supported by the National Natural Science Foundation of China (61971123).

### Author Contributions

C Cheng: software, formal analysis, methodology, and writing—original draft, review, and editing.
P Xiao: conceptualization, supervision, funding acquisition, and writing—review and editing.

### Conflict of Interest Statement

The authors declare that they have no conflict of interest.

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
