## [Reviewer comments · Life Science Alliance]

Life Science Alliance

Evaluation of the correctable decoding sequencing as a new powerful strategy for DNA sequencing

Chu Cheng and Pengfeng Xiao

DOI: <https://doi.org/10.26508/lsa.202101294>

Corresponding author(s): Pengfeng Xiao, Southeast University

Review Timeline:

Submission Date:	2021-11-10
Editorial Decision:	2022-01-17
Revision Received:	2022-03-02
Editorial Decision:	2022-03-31
Revision Received:	2022-04-01
Accepted:	2022-04-01

Scientific Editor: Novella Guidi

Transaction Report:

January 17, 2022

Re: Life Science Alliance manuscript #LSA-2021-01294

Prof. Pengfeng Xiao
Southeast University
Si Pai Lou 2#, Xuanwu District
Nanjing 210096
China

Dear Dr. Xiao,

Thank you for submitting your manuscript entitled "Evaluation of the correctable decoding sequencing as a new powerful strategy for DNA sequencing" to Life Science Alliance. The manuscript was assessed by expert reviewers, whose comments are appended to this letter. We, thus, encourage you to submit a revised version of the manuscript back to LSA that responds to all of the reviewers' points.

Thank you for this interesting contribution to Life Science Alliance. We are looking forward to receiving your revised manuscript.

Sincerely,

B. MANUSCRIPT ORGANIZATION AND FORMATTING:

Reviewer #1 (Comments to the Authors (Required)):

The Manuscript „Evaluation of the correctable decoding sequencing as a new powerful strategy for DNA sequencing" describes a theoretical approach to enhance accuracy of existing next generation sequencing platforms. The innovation it presents consist of replacing the single-base flows in conventional single nucleotide addition sequencing workflows with dual-base additions. This is not entirely new and will result in highly ambiguous sequencing results, that can only be interpreted by repetition of the sequencing workflow with different dual-base combinations. The approach presented here replaces the dual-base addition of two natural nucleotides with a mixture of a natural and a reversibly terminated nucleotide. Combined with regular deblocking flows to remove the terminators This will result in less ambiguous sequencing results that can be decoded by only two consecutive sequencing runs with the same template.

The approach seems original and the claim for higher accuracy is justified. Some simulated results are included. I would support acceptance of the manuscript, after some minor revisions: (1) A software is mentioned that can simulate sequencing results as they would be expected from the described sequencing method. As the ability to correct sequencing errors is claimed, a reasonably large set of sequencing reads containing errors according to the typical error profile of one of the mentioned commercial systems should be generated and subjected to the proposed error correction. The outcome of this simulation should be shown to justify the claim of extraordinary sequence accuracies, eg by presenting a statistics about the raw error rate of the simulated reads and a final error rate after algorithmic error correction. (2) The overall language quality of the manuscript is quite poor with a lot of grammar errors, incomplete sentences, missing words and incorrect expressions. Therefore language editing by a native or highly proficient english speaking person seems necessary.

Reviewer #2 (Comments to the Authors (Required)):

The manuscript "Evaluation of the correctable decoding sequencing technology as a new powerful strategy for DNA sequencing" proposed the new protocol to reduce the error in existing NGS platforms. As there are so many applications of NGS, one of them is pharmacogenomics studies which are mainly based on SNPs. less error in sequencing is directly proportional to less false-positive SNPs and other variations, there is a need for such protocol which fulfill this requirement.

Major Issue

The overall manuscript is in good shape and provides a useful alternative, but, as the author proposed new software to analyze and assemble sequences how they claim that they reduce the shortcoming of NGS platforms. Here as per the manuscript, it seems that they relied on the same platform but proposed the new algorithm/ software to analyze sequence more accurately. They should clearly claim for a new protocol, not a platform.

There are a few minor queries as well

- Author claims single-read information accuracy, what is the basis of this claim?
- What will be standard sequence coverage for the proposed method?

I want to thank all reviewers for giving us constructive suggestions which would help us to improve the quality of the paper. We have made great effort to revise the manuscript in accordance with the reviewers' comments, and carefully proof-read the manuscript to minimize typographical, grammatical, and bibliographical errors. Here below is our point-to-point response to the two reviewers' comments.

Reviewer #1:

Comment 1: A software is mentioned that can simulate sequencing results as they would be expected from the described sequencing method. As the ability to correct sequencing errors is claimed, a reasonably large set of sequencing reads containing errors according to the typical error profile of one of the mentioned commercial systems should be generated and subjected to the proposed error correction. The outcome of this simulation should be shown to justify the claim of extraordinary sequence accuracies, eg by presenting a statistics about the raw error rate of the simulated reads and a final error rate after algorithmic error correction.

Response: Thanks for your valuable comments on the paper. The statistics for the simulation of sequencing under different raw error rate have been added in Result Section "Error correction strategy of the correctable decoding sequencing approach".

Comment 2: The overall language quality of the manuscript is quite poor with a lot of grammar errors, incomplete sentences, missing words and incorrect expressions.

Therefore language editing by a native or highly proficient English speaking person seems necessary.

Response: We are very sorry for our language problem. We have checked the full text and the spelling and grammar errors have been checked and corrected. The level of English throughout the manuscript have been language polished by MJEditor (www.mjeditor.com)

Reviewer #2:

Comment 1: The overall manuscript is in good shape and provides a useful alternative, but, as the author proposed new software to analyze and assemble sequences how they claim that they reduce the shortcoming of NGS platforms. Here as per the manuscript, it seems that they relied on the same platform but proposed the new algorithm/software to analyze sequence more accurately. They should clearly claim for a new protocol, not a platform.

Response: Thank you for the comments on the paper. Your advice is very good. The approach we proposed is a new protocol, not a platform. We have changed in this manuscript.

Comment 2: Author claims single-read information accuracy, what is the basis of this claim?

Response: A template needs to be interrogated twice in the correctable decoding sequencing approach. Therefore, a completely aligned sequence of consecutive bases can be obtained only for two different sequential sequences with the same base deletion or insertion at the same base position (including homopolymer region), as well as with the same pattern and number of sequencing errors. In the simulation, we calculated a conservative theoretical error rate of 0.0009% (lower than that for Sanger sequencing). Thus, this proposed approach can be used to assess the correctness of a single sequencing read.

Comment 3: What will be standard sequence coverage for the proposed method?

Response: Unlike the current NGS, the template is interrogated via multiple parallel sequencing runs (not simple repetitions) with the correctable decoding sequencing method, we can determine whether the original sequencing information is correct by aligning any two sets of sequencing information, and the corrected "alignment information" can be used as a read for sequence assembly. Therefore, theoretically, accurate sequence information can be obtained when the sequencing depth is 1×, thereby reducing the coverage required for assembly. In general, the sequencing depth refers to the coverage depth of the sequencing data on the genome. Coverage is the proportion of the genome that can be covered by sequencing data compared to the reference genome. For example, if the human genome is 3G, we measured and filtered 90G of clean data, then the sequencing depth is $90/3=30x$, and the coverage is 80%. Therefore, the specific coverage is related to the composition of the genome to be sequenced and fragmentation technology.

We appreciate for Editors/Reviewers' warm work earnestly, and hope that the correction will meet with approval. Should you have any questions, please contact us without hesitate.

Once again, thank you very much for your comments and suggestion.

Sincerely yours,

Corresponding author:

Name: Pengfeng Xiao

E-mail: xiaopf@seu.edu.cn

March 31, 2022

RE: Life Science Alliance Manuscript #LSA-2021-01294R

Prof. Pengfeng Xiao
Southeast University
Si Pai Lou 2#, Xuanwu District
Nanjing 210096
China

Dear Dr. Xiao,

Thank you for submitting your revised manuscript entitled "Evaluation of the correctable decoding sequencing as a new powerful strategy for DNA sequencing". We would be happy to publish your paper in Life Science Alliance pending final revisions necessary to meet our formatting guidelines.

- please add ORCID ID for the corresponding author-you should have received instructions on how to do so
- please add the Twitter handle of your host institute/organization as well as your own or/and one of the authors in our system
- titles in the system and manuscript file must match
- please consult our manuscript preparation guidelines <https://www.life-science-alliance.org/manuscript-prep> and make sure your manuscript sections are in the correct order;
- please add an Author Contributions section to your main manuscript text
- tables should be included at the bottom of the main manuscript file or be sent as separate files
- please add your main, supplementary figure, and table legends to the main manuscript text after the references section
- please label the last figure as figure 6 (currently there are two figures labeled as figure 5)
- please provide a Data Availability section

A. FINAL FILES:

B. MANUSCRIPT ORGANIZATION AND FORMATTING:

Sincerely,

Reviewer #1 (Comments to the Authors (Required)):

The revised version of the manuscript „Evaluation of the correctable decoding sequencing as a new powerful strategy for DNA sequencing" describes a theoretical approach to enhance accuracy of existing next generation sequencing platforms. The innovation it presents consist of replacing the single-base flows in conventional single nucleotide addition sequencing workflows with dual-base additions. This is not entirely new and will result in highly ambiguous sequencing results, that can only be interpreted by repetition of the sequencing workflow with different dual-base combinations. The approach presented here replaces the dual-base addition of two natural nucleotides with a mixture of a natural and a reversibly terminated nucleotide. Combined with regular deblocking flows to remove the terminators. This will result in less ambiguous sequencing results that can be decoded by only two consecutive sequencing runs with the same template.

The points raised during the first review have now been adequately treated and can in the present form be recommended for publication

Reviewer #2 (Comments to the Authors (Required)):

The authors include all the suggested points, and now the manuscripts seem in a good and acceptable format.

April 1, 2022

RE: Life Science Alliance Manuscript #LSA-2021-01294RR

Prof. Pengfeng Xiao
Southeast University
Si Pai Lou 2#, Xuanwu District
Nanjing 210096
China

Dear Dr. Xiao,

Thank you for submitting your Methods entitled "Evaluation of the correctable decoding sequencing as a new powerful strategy for DNA sequencing". It is a pleasure to let you know that your manuscript is now accepted for publication in Life Science Alliance. Congratulations on this interesting work.

DISTRIBUTION OF MATERIALS:

Again, congratulations on a very nice paper. I hope you found the review process to be constructive and are pleased with how the manuscript was handled editorially. We look forward to future exciting submissions from your lab.

Sincerely,
